# Double-Gated Mamba Multi-Scale Adaptive Feature Learning Network for Unsupervised Single RGB Image Hyperspectral Image Reconstruction

**DOI:** 10.3390/jimaging12010019

**Published:** 2025-12-31

**Authors:** Zhongmin Jiang, Zhen Wang, Wenju Wang, Jifan Zhu

**Affiliations:** College of Publishing, University of Shanghai for Science and Technology, Shanghai 200093, China; jzmn@usst.edu.cn (Z.J.); wangwenju@usst.edu.cn (W.W.); 243362808@st.usst.edu.cn (J.Z.)

**Keywords:** RGB image, HSI reconstruction, Mamba, multi-scale adaptive feature learning

## Abstract

Existing methods for reconstructing hyperspectral images from single RGB images struggle to obtain a large number of labeled RGB-HSI paired images. These methods face issues such as detail loss, insufficient robustness, low reconstruction accuracy, and the difficulty of balancing the spatial–spectral trade-off. To address these challenges, a Double-Gated Mamba Multi-Scale Adaptive Feature (DMMAF) learning network model is proposed. DMMAF designs a reflection dot-product adaptive dual-noise-aware feature extraction method, which is used to supplement edge detail information in spectral images and improve robustness. DMMAF also constructs a deformable attention-based global feature extraction method and a double-gated Mamba local feature extraction approach, enhancing the interaction between local and global information during the reconstruction process, thereby improving image accuracy. Meanwhile, DMMAF introduces a structure-aware smooth loss function, which, by combining smoothing, curvature, and attention supervision losses, effectively resolves the spatial–spectral resolution balance problem. This network model is applied to three datasets—NTIRE 2020, Harvard, and CAVE—achieving state-of-the-art unsupervised reconstruction performance compared to existing advanced algorithms. Experiments on the NTIRE 2020, Harvard, and CAVE datasets demonstrate that this model achieves state-of-the-art unsupervised reconstruction performance. On the NTIRE 2020 dataset, our method attains MRAE, RMSE, and PSNR values of 0.133, 0.040, and 31.314, respectively. On the Harvard dataset, it achieves RMSE and PSNR values of 0.025 and 34.955, respectively, while on the CAVE dataset, it achieves RMSE and PSNR values of 0.041 and 30.983, respectively.

## 1. Introduction

The spectral resolution of hyperspectral images (HSIs) is on the order of 
$10^{-2}\lambda$
. The amount and resolution of spectral information they contain are far greater than those of multispectral and RGB images. HSIs have been widely used in image information processing fields, particularly in remote sensing and Earth observation [1], agriculture and crop monitoring [2], mineral exploration and identification [3], environmental pollution detection [4], and cultural heritage preservation and archeology [5]. However, the acquisition of hyperspectral images requires expensive sensor equipment and technology, large-capacity data storage and transmission needs, as well as specialized knowledge and resources for the complex data collection and processing procedures, which severely limits their widespread application.

To overcome these limitations and considering the advantages of RGB images, such as low acquisition cost, high spatial resolution, and rich texture details, an increasing number of researchers have focused on recovering hyperspectral images (HSIs) from RGB images by learning the dependencies and correlations between RGB images, a process known as spectral reconstruction. Methods for hyperspectral image reconstruction based on RGB images can supplement spectral dimensional information, improve image processing performance, and reduce hardware costs and complexity, making them a research hotspot in the fields of image processing and computer vision. Early works [6,7,8] utilized prior knowledge such as low-rankness and sparsity to crudely model shallow feature representations. However, due to the simplicity of model design and over-reliance on prior information, the accuracy and generative capability of these methods were limited.

With the advantages of automatic feature learning, high performance, and scalability, deep learning models have been widely applied in fields such as computer vision [9], natural language processing [10], and medical image analysis [11], achieving remarkable results. Given the excellent performance of deep learning, researchers have started to apply deep learning techniques to the field of hyperspectral image reconstruction from a single RGB image. Based on the different neural network architectures in deep learning, the reconstruction methods can be categorized into CNN-based methods [12,13], GAN-based methods [14,15], Attention-based methods [16,17], and Transformer-based methods [18,19]. Most CNN-based methods, while optimizing pixel-level distances between the generated and real images, tend to over-smooth the images, affecting the diversity and authenticity of the spectra. GAN-based spectral imaging methods, to avoid instability during training, typically require the exclusion of training data that do not match the test set, leading to poor model robustness. Attention-based reconstruction methods enhance feature extraction by deepening the network, expanding the network scale, and integrating multiple networks, which improves reconstruction performance, but this significantly reduces the speed of spectral image reconstruction. Although Transformer-based methods can account for the spatial sparsity of spectral information, they fail to effectively model spectral similarity and long-range dependencies, and maintaining a balance between spatial and spectral resolution remains an unresolved issue.

To address these problems, this paper proposes a Dual-Gated Mamba Multi-Scale Adaptive Feature Learning (DMMAF) network, which can reconstruct high-precision hyperspectral images from a single RGB image. Specifically, our network adopts a reflection dot-product adaptive dual-noise-aware feature extraction method, which effectively supplements the edge detail information of the spectral images and enhances the model’s robustness. Furthermore, DMMAF introduces a deformable attention-based global feature extraction and dual-gated Mamba local feature extraction method, which strengthens the interaction between local and global information, thereby improving the accuracy of image reconstruction. Meanwhile, DMMAF proposes a structure-aware smooth loss function, which successfully addresses the challenge of balancing spatial and spectral resolution by integrating smooth loss, curvature loss, and attention supervision loss.

The main contributions of our work are summarized as follows:(1)We propose the Double-Gated Mamba Multi-Scale Adaptive Feature (DMMAF) learning network for high-precision reconstruction from RGB to HSIs. The network primarily consists of three components: Reflection Dot-product Adaptive Dual-noise-aware Feature Extraction, Deformable Attention Dual-gated Mamba Multi-Scale Feature Learning, and Structure-aware Smooth Constraint Loss Function. Extensive experimental results demonstrate that this algorithm can reconstruct a high-precision HSI from a single RGB image, outperforming other advanced reconstruction algorithms during unsupervised training.(2)DMMAF designs a Reflection Dot-product Adaptive Dual-noise-aware Feature Extraction method. This method is primarily composed of two reflection depth dot-product feature processing modules and an adaptive dual-noise mask module. The reflection depth dot-product module primarily implements channel transformation and supplements edge spatial information details, while the adaptive dual-noise mask enhances feature bandwidth correlation and contextual relationships, thereby improving the model’s robustness.(3)DMMAF constructs a Deformable Attention Dual-gated Mamba Multi-Scale Feature Learning method. The deformable attention modeling addresses the issue of insufficient attention to important information in global features during image reconstruction, thereby mitigating the problem of detail loss. The Dual-gated Mamba local feature extraction resolves the global–local feature conflict and reduces channel redundancy, resulting in a significant reduction in parameters. The combination of these two methods enhances the interaction efficiency between local and global information during image reconstruction, thus improving the accuracy of the reconstruction.(4)DMMAF introduces a structure-aware smooth loss function. This module comprises three loss functions: smooth loss, curvature loss, and attention supervision loss, which address issues such as the neglect of image structural information, a lack of spatial structural constraints, and insufficient supervision of the attention mechanism. It effectively balances spatial and spectral resolution.

## 2. Related Work

Hyperspectral image reconstruction from a single RGB image based on deep learning can generally be categorized into three types based on different neural network architectures: supervised reconstruction, semi-supervised reconstruction, and unsupervised reconstruction.

In supervised RGB-to-hyperspectral image reconstruction, the model is trained using a large dataset of paired RGB and hyperspectral images [20], where the RGB image is used to predict and reconstruct the hyperspectral image. The Attention-based Scale Feature-Adversarial Network (SAPUNet) [16] combines attention mechanisms (scale attention pyramid UNet) and feature pyramids (scale attention pyramid W-Net) to reconstruct hyperspectral images from RGB images. This network generates results with spatial consistency and less blurring by combining content loss and adversarial loss; however, it fails to reconstruct high-frequency details. The RGB-to-Hyperspectral Generative Adversarial Network (R2HGAN) [15] utilizes a conditional discriminator and spectral discriminator joint discriminative method, showing great realism in spectral reconstruction. However, this model requires reassembling the RGB image, which leads to inefficiencies in memory usage. The hyperspectral image reconstruction model based on deep convolutional neural networks (HSCNN-R) [12] learns the mapping relationship between RGB and hyperspectral images through a training dataset and optimizes the model parameters using a loss function. However, this model suffers from a significant amount of feature overlap, increasing computation time. The HSGAN network introduces a two-stage adversarial training strategy and a spatial–spectral attention mechanism for RGB-to-hyperspectral image reconstruction [14], improving denoising capabilities and enhancing feature representation. However, its generalization ability across different RGB images and scenes is poor, resulting in unstable performance. The Adaptive Weighted Attention Network (AWAN) [21], consisting of 12 dual-residual attention blocks (DRABs) and a PSNL module, exhibits excellent adaptability to scene variation and can reconstruct high accuracy and visual effects when constrained by input illumination data. However, its reconstruction accuracy deteriorates under illumination changes. The Dual Hierarchical Regression Network (DHRNet) [22] designs a shadow feature extraction module based on a dense structure and a reflection feature extraction module with an attention mechanism to reconstruct spectral information from reflection and shadow features in the presence of illumination variation. This network suppresses spectral distortion and enhances the clarity of hyperspectral image reconstruction. However, the attention mechanism overly focuses on local regions, impacting the accuracy of spectral reconstruction. The Correlation and Continuity Network (CCNet) [23] solves the challenge of multi-scale fusion in RGB-to-hyperspectral reconstruction by designing spectral correlation modeling and neighborhood spectral continuity modules that balance local and global feature similarity and continuity. It also incorporates an adaptive fusion module to improve the complementarity between modules. However, this method overlooks the global context and non-local correlations of features, leading to a decline in reconstruction quality. The Wavelet-based Dual Transformer Model (WDTM) [24] combines dual attention mechanisms with wavelet signal decomposition, capturing non-local spatial similarity and global spectral correlation, while improving computational efficiency. However, its robustness under different imaging conditions is limited. Supervised learning for RGB-to-hyperspectral reconstruction can better learn the mapping relationship between paired RGB and hyperspectral data, achieving high spectral precision and spatial resolution in reconstruction, making it suitable for complex scenes. However, this approach requires a large amount of accurately labeled RGB-HSI paired data, which is costly, and the reconstruction of hyperspectral images requires complex model structures and high computational resources, which can lead to model overfitting.

In hyperspectral image reconstruction, semi-supervised methods use a small amount of paired RGB and hyperspectral images, along with a large amount of unlabeled RGB images, reducing the computational resource and cost issues associated with supervised learning. Semi-supervised deep learning techniques [25] employ RGB space reverse mapping to form a total variation regularized model for training a limited number of hyperspectral images, improving reconstruction accuracy. However, this method does not incorporate adversarial learning techniques, and the limited labeled data results in low robustness. Such methods can adapt to scene changes and improve the accuracy of hyperspectral image reconstruction by using a small amount of labeled data and a large amount of unlabeled data. However, they suffer from limitations due to assumptions about the data, and unlabeled data may contain noise. Additionally, insufficient labeled data can hinder the effective learning of complex features, impacting reconstruction performance.

Unsupervised RGB-to-hyperspectral image reconstruction does not rely on large amounts of manually labeled hyperspectral data, enabling training even when labeled data are scarce. The class-based backpropagation neural network (BPNN) [26] uses an unsupervised clustering method to partition RGB and hyperspectral images into several pairs for training, and the trained BPNN reconstructs the final hyperspectral image from the classified RGB images. However, the BPNN algorithm has slow learning speeds and is prone to training failures. The Deep Residual Convolutional Neural Network (DRCNN) model [13] similarly uses an unsupervised clustering method to establish a nonlinear spectral mapping between RGB and hyperspectral image pairs, demonstrating significant applicability and effectiveness. However, the proposed method requires training multiple DRCNNs, leading to inefficient networks and reduced reconstruction accuracy. A perceptual imaging degradation model using the camera spectral response function effectively addresses the low efficiency of single RGB-to-hyperspectral reconstruction [27] while achieving high reconstruction accuracy. However, the model is limited by unknown RGB image degradation conditions in real-world settings, reducing the diversity of imaging. To mitigate the dependency of all HSI-SR models on RGB image degradation conditions, the camera response function (CRF) is used for RGB-to-hyperspectral image reconstruction [28]. However, this model is constrained by the camera sensor, which can cause spectral distortion in RGB images. The Random Resonance Iterative Model based on Two Types of Illumination [29] accurately approximates the gradient algorithm to solve the resonance iterative model, significantly improving both visual and quantitative evaluations. However, the algorithm’s solution process is complex, reducing the model’s robustness. The Deep Low-Rank Hyperspectral Image model (DLRHyIn) [30] uses ℓ2 norm squared as the data fitting function, applying unsupervised high-order algorithms (DIP) and smooth low-rank regularization to improve network convergence speed. However, the method’s effectiveness is not guaranteed in the presence of outliers or sparse noise. The SkyGAN model based on the GAN framework [31] combines adversarial distribution difference alignment and cycle consistency constraints in an unsupervised, unpaired manner to reconstruct domain-aware hyperspectral images from natural scene RGB, reducing training cycles. However, this model is susceptible to collapse during the learning process, and the generator may degrade, losing image details. The Unsupervised Spectral Super-Resolution Decomposition Guided Unsupervised Network (UnGUN) [32] enables single RGB-to-hyperspectral reconstruction without paired images. This network includes two decomposition branches for RGB and hyperspectral images, along with a comprehensive reconstruction branch, ensuring the reconstructed image follows the features of real hyperspectral images. However, the decomposition and reconstruction processes require module adjustments and discriminator support, making the algorithm cumbersome. The Masked Transformer (MFormer) network [18] uses a dual-frequency spectral self-attention (DSSA) module and a Multi-Head Attention Block (MAB) module for hyperspectral image reconstruction, capturing fine spectral features while enhancing network generalization and effectiveness. However, the masking reconstruction operation is computationally intensive and suffers from challenges in balancing spatial–spectral resolution.

As seen, compared to supervised and semi-supervised methods, unsupervised single RGB-to-hyperspectral image reconstruction offers advantages such as lower data labeling and collection costs, stronger generalization ability, and higher data utilization efficiency. However, it still faces challenges such as loss of detail, insufficient robustness, low reconstruction accuracy, and difficulty in balancing spatial and spectral resolution.

## 3. Methodology

A single RGB image is used as the input to the DMMAF model to obtain the hyperspectral image reconstructed under the unsupervised method. The DMMAF network framework is illustrated in Figure 1. It consists of three main parts: (a) Reflection Dot-Product Adaptive Dual-noise-Aware Feature Extraction (RDPADN), (b) Deformable Attention Dual-Gated Mamba Multi-Scale Feature Learning (DADGM), and (c) Structure-Aware Smooth Constraint Loss Function. RDPADN is primarily used for fine feature edge extraction and adaptive noise processing of a single RGB image (Section 3.1). This module is composed of two Reflection Depth Point Feature Extraction (RDPFE) modules and an Adaptive Dual-Noise Masking (ADNM) module. It uses a reflection dot-product mechanism to highlight structural boundaries and suppress noise, allowing the model to retain fine spatial details during the subsequent reconstruction process. DADGM mainly consists of the dual encoding structure Attention-Mamba Layer (AMLayer), the Double-Mamba Layer (DMLayer), and the decoding structure Attention-Feedforward Layer (AFLayer). DADGM accepts refined features from the RGB image and, based on the even-odd indexing of the encoder layers, enters the AMLayer and DMLayer to perform multi-scale feature downsampling and upsampling. Finally, the features are fused into the AFLayer to achieve local and global association modeling (Section 3.2). The Structure-Aware Smooth Constraint Loss Function module calculates the loss function according to the number of iterations (Section 3.3). This loss function encourages spatial smoothness, maintains structural boundaries, and enhances spectral consistency without relying on paired RGB-HSI labels. If the preset epoch number is not reached, the loss calculation returns to the encoder index layer for further processing through the dual-branch encoder, creating a cyclic pattern to optimize model performance and improve reconstruction accuracy. Once the epoch number is met, the final reconstructed hyperspectral image is outputted.

### 3.1. Reflection Dot-Product Adaptive Dual-Noise-Aware Feature Extraction

In the hyperspectral image reconstruction task, feature extraction from the RGB image is a crucial step that enables the model to recover key spectral channels ranging from dozens to hundreds using limited visible light information. This feature extraction forms the foundation for the effectiveness of the entire reconstruction task. However, existing feature extraction methods suffer from issues such as high convolution overhead, loss of edge information, coupling of spatial and channel information, and weak representation of key regional features [33,34]. To address these challenges, this paper proposes a Reflection Dot-product Adaptive Dual-noise-aware Feature Extraction method.

The Reflection Dot-Product Adaptive Dual-noise-Aware Feature Extraction method consists of two Reflection Depth Point Feature Extraction (RDPFE) modules and an Adaptive Dual-Noise Masking (ADNM) module. The main implementation process is shown in Figure 2. The features of the RGB image are input into the Reflection Depth Point Feature Extraction (RDPFE) submodule to obtain shallow features 
$F_{out0}$
, At this stage, the feature map channels are expanded from 3 dimensions to 31 dimensions. The shallow feature 
$F_{out0}$
 are processed through the Adaptive Dual-Noise Masking (ADNM) module to generate adaptive mask features 
$F_{Mask}$
, with the number of channels remaining unchanged. 
$F_{Mask}$
 is then passed through the RDPFE module to extract multi-channel refined features 
$F_{out1}$
. Clearly, the method primarily involves the reflection depth dot-product and the adaptive dual-noise masking components.

#### 3.1.1. Reflection Depth Point Feature Extraction

The foundation of single RGB image reconstruction of hyperspectral images lies in the extraction of features from the RGB image. This is primarily achieved using Convolutional Neural Networks (CNNs) to extract local spatial features or by introducing Transformer models to model long-range dependencies and capture global contextual information. However, CNN models [35] struggle to capture spectral dependencies across channels, while Transformer-based models incur high computational costs [36], leading to overfitting or training instability. To address these challenges, this paper proposes Reflection Depth Point Feature Extraction (RDPFE), which enhances the incorporation of edge spatial information using reflection padding and depth-wise separable convolution techniques. This method effectively resolves spectral dependencies across channels while reducing computational overhead.

Reflection Depth Point Feature Extraction (RDPFE) is primarily utilized for feature extraction and channel dimension transformation in images. It mainly consists of reflection padding, depth-wise convolution, and point-wise convolution processes, as illustrated in Figure 3.

(1)Reflection Padding

In image processing tasks, edge pixels often contain important information such as object contours. Reflection padding extends the boundaries by symmetrically copying the edge pixels of the input tensor before convolution, while dynamically adjusting based on the dilation rate of the convolution to reduce boundary artifacts introduced by the convolution. Specifically, after preprocessing a single RGB image, the resulting feature map is subjected to reflection padding 
$X\in R^{H\times W\times 3}$
, the features 
$F_{Pad}$
 can be represented by Equation (1).
(1)
$$F_{Pad}=Pad_{reflect}(X,p)$$
In this case, 
p=d·(k−1)/2
 represents the padding width, 
k
 is the size of the padding convolution kernel, and 
d
 is the dilation coefficient. The padding operation does not introduce additional invalid values (such as 0); instead, it utilizes the reflective information of the edge pixels, effectively reducing the generation of edge artifacts in the image.

(2)Depth Convolution

The deep convolution independently convolves the reflection-padded features 
$F_{pad}$
 across the input channels, with a kernel size of 
3×3
. The processed three-dimensional feature 
$F_{dw}$
 can be represented by the following formula:
(2)
$$F_{dw}=\sum_{i=1}^{3}Conv3\times 3(F_{Pad})$$


Compared to traditional fully connected convolutions, depth-wise convolution significantly improves computational efficiency while retaining spatial features, making it suitable for constructing lightweight models.

(3)Point-Wise Convolution

Point-wise convolution uses 31 1 × 1 convolution kernels to simultaneously perform convolution operations across the channels of feature 
$F_{dw}$
 to obtain the output features 
$F_{\mathrm{out}0}$
, as shown in Equation (3).
(3)
$$F_{\mathrm{out}0}=\sum_{i=1}^{31}C\mathrm{onv}1\times 1(F_{dw})$$


The point-wise convolution linearly combines all input channels at each pixel position of the deep convolution’s output.

#### 3.1.2. Adaptive Dual-Noise Masking

The masking operation enhances the bandwidth correlation of features and the contextual relationships between features, thereby extracting deeper features from the image. Traditional masking modules, such as MAE (Masked Autoencoders) [34] work by masking a portion of the image patches and utilizing an encoder–decoder structure to learn and reconstruct the masked parts. However, traditional MAE is often static, lacking noise-awareness and context adaptation capabilities, resulting in suboptimal efficiency when processing complex hyperspectral images. To address this issue, ADNM is proposed. It primarily includes three core steps: spatial masking, channel perturbation, and adaptive masking, as shown in Figure 4.

(1)Spatial Masking

➀ The spatial masking module receives the features 
$F_{\mathrm{out}0}$
 initially processed by RDPFE. The size of its feature map is 
b×c×h×w
, where 
b
 is the batch size, 
c
 is the number of channels, 
h
 and 
w
 are the height and width of the feature map. The features 
$F_{out0}$
 are then flattened into vectors 
L=c×h×w
, followed by a reshape operation to derive the feature 
$F_{R}\in (b,L)$
 as expressed in Equation (4).
(4)
$$F_{R}=Reshape(F_{out0})$$


➁ A random matrix 
$\varepsilon_{b,L}$
 is generated at each pixel position of the feature 
$F_{R}$
, as shown in Equation (5).
(5)
$$\varepsilon_{b,L}=rand(F_{R})$$
The random matrix 
$\varepsilon_{b,L}$
 assigns a random value to each pixel, and the size of the random value is compared with 
1−p
 to classify and obtain a new binary mask matrix 
$M^{\left(i,j\right)}$
, as shown in the following formula:
(6)
$$M^{\left(i,j\right)}=\left\{\begin{array}{l} 0\;\varepsilon_{b,L}\leq 1-p\\ 1\;otherwise \end{array}\right.$$

where 
(i,j)
 represent the pixel position in the 
i
-th column and 
j
-th row. When the random value is greater than 
1−p
, it is masked; otherwise, it is retained.

➂ In the binary mask matrix 
M
, the pixel positions where the matrix value is 1 introduce noise 
η
, which acts on 
$F_{R}$
 to obtain the spatial mask feature 
$F_{s}$
 as shown in Equation (7):
(7)
$$F_{s}=F_{R}+M\cdot 0.1\eta$$

where 0.1 represents the intensity of the noise 
η
.

(2)Channel Perturbation

Building upon the spatial position random masking, the ADNM module further introduces a channel-level masking strategy to perform channel perturbation. A certain proportion of channels are selected from 
$F_{s}$
, and noise is added to all pixel values of these channels. 
r
 represents the proportion of randomly selected channels, σ denotes the noise intensity, and 
N(0,1)
 is generated from a standard normal distribution. Therefore, after the noise is added, the channel perturbed feature 
$F_{P}$
 can be expressed as shown in Equation (8).
(8)
$$F_{P}=Select(F_{s}\cdot r)+\sigma \cdot N(0,1)$$


(3)Adaptive Masking

The model dynamically updates the spatial and channel masking ratios 
ρ
 and 
r
 based on the number of iterations, thereby repeating the following steps ➀ and ➁, until the set epoch number is reached, at which point 
$F_{Mask}$
 is output. Therefore, within the iterative loop, the adaptive masking module controls the proportion of features to be masked in the spatial masking module and the number of channels to be masked in the channel perturbation. The overall masking ratio of the ADNM module increases linearly, preserving more features in the early stages of training and gradually increasing the masking intensity in the later stages. This approach helps improve the model’s stability and convergence speed under high masking ratios.

➀ The iterative update of the spatial masking ratio 
p
 is generally as follows: the value of 
p
 increases with the number of training iterations. It is represented by Equation (9):
(9)
$$p=p_{0}+\alpha_{1}\cdot \frac{N_{t}}{N_{T}}$$


Here, 
$p_{0}$
 is the initial masking ratio, set to 0.3, and 
$\alpha_{1}$
 is the adjustment coefficient, with a default value of 0.5. 
$N_{t}$
 represents the current training epoch, and 
$N_{T}$
 is the total number of training epochs.

➁ The iterative update of the channel masking ratio 
r
 is generally as follows: the value of 
r
 increases with the number of training iterations, as represented by Equation (10).
(10)
$$r=r_{0}+\alpha_{2}\cdot \frac{N_{t}}{N_{T}}$$


In this case, 
$r_{0}$
 is the initial masking ratio, set to 0.2, and 
$\alpha_{2}$
 is the adjustment coefficient, with a default value of 0.2.

### 3.2. Deformable Attention Dual-Gated Mamba Multi-Scale Feature Learning

To address the issues of insufficient long-range dependency modeling, inefficient multi-scale feature fusion, and low interaction efficiency between local and global information in traditional CNNs and single Transformer architectures, a Deformable Attention Dual-Gated Mamba Multi-Scale Feature Learning method has been designed. The process of this method is illustrated in Figure 5. The multi-channel refined feature 
$F_{out1}$
 is divided into two types of features, 
$F_{D}$
 and 
$F_{S}$
, based on the odd–even indexing of the stacked encoder layers, and they are processed separately by the AMLayer and DMLayer.

The distinction between these two processing structures lies in the use of different encoders: Encoder A in the AMLayer employs a combination of the attention mechanism and Mamba processing, while Encoder B in the DMLayer utilizes the dual Mamba processing method. In the AMLayer, the feature 
$F_{out1}$
 is constrained by the Coder Depth indexing layer to form the even-layer feature 
$F_{D}$
, which then enters Encoder A and undergoes the first downsampling operation via the SElayer. The downsampling result subsequently enters Encoder A for a second processing, followed by the second downsampling and the first upsampling operations. The upsampling result is then processed by Encoder A for the third time, followed by the second upsampling, yielding the final result 
$F_{1}$
. The processing procedure in the DMLayer is identical to that of the AMLayer. Both AMLayer and DMLayer processing paths involve down-down-up-up sampling. To better illustrate their implementation, we have provided the pseudocode for the sampling path algorithm in Table A1. The processed feature 
${F_{1}}^{\prime }$
 from this layer is fused with 
$F_{1}$
 to generate 
$F_{C}$
. This fused feature then enters the RDPFE for fine-tuning of the image features, producing 
$F_{out2}$
. Subsequently, it passes through the two decoders that form the AFLayer (Attention-Feedforward Layer), and then re-enters the RDPFE (as described in Section 3.1.1) to output the feature 
$F_{out3}$
.

The Encoder and Decoder of the AMLayer, DMLayer, and AFLayer layers are distinguished by their internal structural compositions, and are denoted as Encoder A, Encoder B, and Decoder, as shown in Figure 6. In Encoder A, the image feature 
$F_{D_{i}}$
 undergoes Prenorm normalization and is then processed through the Deformable Attention (DA) submodule and the Dual Gated Mamba (DGM) submodule for dynamic global feature modeling, local feature modeling, and fusion. The fused feature is processed by the Feedforward Network (FFN) then added to the input feature 
$F_{D_{i}}$
 to perform a residual connection before being output. In Encoder B, 
$F_{S_{i}}$
 undergoes Prenorm normalization, passes through two DGM submodules, is concatenated, and is then processed through the FFN. The output is then combined with the original feature 
$F_{S_{i}}$
 through a residual connection. In the Decoder, the feature 
$F_{out2_{i}}$
 also undergoes Prenorm normalization, passes through two DA submodules, is concatenated, and is then processed through the FFN. The output is then combined with the original feature 
$F_{out2_{i}}$
 through a residual connection. It can be seen that DA and DGM are the core and critical modules in both the encoder and decoder; therefore, a detailed introduction to these two modules is necessary in the following sections.

#### 3.2.1. Deformable Attention Global Feature Extraction

The Deformable Attention (DA) submodule dynamically allocates weights to different parts of the input, enabling focused attention on important information and the effective integration of contextual details. The structure of the DA module is illustrated in Figure 7, and its specific technical details can be divided into the following five steps:

➀ Construct Multi-head Vectors. The feature 
$F_{D_{i}}$
 undergoes a linear transformation through a linear layer to construct multi-head query vectors 
Q
, key vectors 
K
, and value vectors 
V
, as shown in Equation (11).
(11)
$$Q,K,V=Linear(F_{D_{i}})$$
 where 
$Q,K,V\in R^{b\times n\times (h\cdot d)}$
, 
b
 is the batch size, 
n
 is the total number of spatial positions, 
h
 is the number of attention heads, 
d
 is the dimensionality of each head.

➁ Learn spatial offsets and generate sampling grids. The feature 
$F_{D_{i}}$
 undergoes a Conv2d convolution operation to predict the two-dimensional offset for each spatial position corresponding to its multi-head attention. The offset 
$\Delta \in R^{b\times 2h\times h\times w}$
 can be expressed as shown in Equation (12).
(12)
$$\Delta =Conv2d(F_{D_{i}})$$


The base sampling grid coordinates 
$G\in {\left[-1,1\right]}^{h\times w\times 2}$
 are updated by adding the spatial offset 
Δ
 resulting in the sampling grid 
$G^{\prime }$
, as shown in Equation (13).
(13)
$$G^{\prime }=G+\Delta$$


➂ Key-value sampling of the sampling grid. K and V are based on the constructed sampling grid 
$G^{\prime }$
 bilinear interpolation (BI) is used for resampling to obtain 
$\tilde{K}$
 and 
$\tilde{V}$
, as shown in Equations (14) and (15).
(14)
$$\tilde{K}=BI(K,G^{\prime })$$

(15)
$$\tilde{V}=BI(V,G^{\prime })$$


➃ Attention weight calculation. The transpose of 
Q
 and 
$\tilde{K}$
 is processed using Softmax to obtain the attention weights 
$A\in R^{b\times h\times n\times n}$
, as shown in Equation (16).

(16)
$$A=soft\max(\frac{Q{\tilde{K}}^{T}}{\sqrt{d}})$$


➄ Attention-weighted output. The product of the attention weights 
A
 and the sampled 
$\tilde{V}$
 values results in the feature 
$Z\in R^{b\times h\times (h\times d)}$
, as shown in Equation (17).
(17)
$$Z=A\tilde{V}$$


The feature 
Z
, after undergoing Dropout processing, is passed through Layer Scale and a projection layer to obtain the final output feature 
$F_{DA}$
, as shown in Equation (18).
(18)
$$F_{DA}=LayerScale(Proj(Dropout(Z)))$$


#### 3.2.2. Dual-Gated Mamba Local Feature Extraction

Due to the high computational and memory overhead, as well as the lack of precise control over long-range dependency modeling in single attention feature extraction mechanisms, a local precise feature modeling method based on a gated state-space model is proposed, as shown in Figure 8.

The specific details of the Dual-Gated Mamba module consist of the following three key steps.

➀ Input-level Gated Adjustment. This operation applies dynamic channel-level weighting to the input features, suppressing noise or irrelevant features and highlighting important information. The input 
$F_{S_{i}}$
 is flattened into a sequential format 
$F_{f}=Flatten(F_{S_{i}})$
, which is then mapped through the linear layer 
Linear()
 to obtain the projection matrices 
$W_{V}$
 and 
$W_{G_{1}}$
. The projection feature values 
V
 are obtained by multiplying 
$W_{V}$
 with 
$F_{f}$
, as shown in Equation (19).
(19)
$$V=W_{V}\cdot F_{f}$$


The sequence length of 
$F_{f}$
 is equal to the height × width of the image, with each pixel becoming a feature sequence of one time step. In this way, the two-dimensional spatial information is linearized into a one-dimensional feature sequence for temporal modeling.

After 
$W_{G_{1}}$
 is multiplied by 
$F_{f}$
, the spatial gated value 
$G_{1}$
 is generated through the SiLU function, as shown in Equation (20).
(20)
$$G_{1}=SiLU(W_{G_{1}}\cdot F_{f})$$


The projection feature value 
V
 is element-wise multiplied by the spatial gated value

$G_{1}$
 to produce the input-level gated conditional feature

$F_{proj}$
 as shown in Equation (21).
(21)
$$F_{proj}=V⊙G_{1}$$


This operation will apply weighting to 
V
. As shown in Equation (21), the larger the value of 
$G_{1}$
, the more features of the projection feature 
V
 are preserved during dot-product operation, meaning the value of 
$F_{proj}$
 becomes larger. Conversely, if 
$G_{1}$
 tends to 0, the value of 
$F_{proj}$
 becomes smaller.

➁ Mamba core discretized state convolution. The input-level gated feature 
$F_{\mathrm{proj}}$
 is transformed into a convolutional format, followed by padding to obtain the feature 
$F_{pad}$
, as shown in Equation (22).
(22)
$$F_{pad}=pad(Conv(F_{proj}))$$


After padding, the feature 
$F_{pad}$
 undergoes a 1d convolution on each channel

$C_{i}$
 resulting in the feature

$F_{1D}$
, as shown in Equation (23).

(23)
$$F_{1D}=Conv1D(F_{Pad},C_{i})$$


State size refers to the feature dimension that the model processes at each time step (or position). In DGM, the feature vector of each pixel has a dimension of 
C
 (the number of channels). After the flatten operation, the model’s state size becomes 
C
, where each pixel position has a feature vector of size 
C


In the state convolution (SSM), multiple convolution kernels 
K
 are element-wise multiplied with the feature 
$F_{1D}$
, resulting in a weighted sum. The final feature 
$F_{SSM}$
 is given by Equation (24).
(24)
$$F_{SSM}=\sum_{k=0}^{L-1}F_{1D}\cdot K_{k}$$


➂ Output-level gated adjustment. The input feature 
$F_{out1}$
 undergoes LayNorm normalization to obtain 
$F_{Norm}$
, which is then passed through a linear layer to generate the gated projection matrix 
$W_{G_{2}}$
. The product of these two is activated by the SiLU function to produce the independent branch channel gated value 
$G_{2}$
 as shown in Equation (25).
(25)
$$G_{2}=SiLU(W_{G_{2}}\cdot F_{Norm})$$



$G_{2}$
 is element-wise multiplied with the feature

$F_{SSM}$
, and after undergoing the dropout operation, the final output

$F_{DGM}$
 is obtained, as represented by Equation (26).

(26)
$$F_{DGM}=Dropout(F_{SSM}⊙G_{2})$$


The channel gate value 
$G_{2}$
 and the state space convolution feature 
$F_{SSM}$
 collaboratively adjust to generate 
$F_{DGM}$


As shown in Equations (19)–(25), 
$G_{1}$
 and 
$G_{2}$
 modulate the input features 
$F_{S_{i}}$
 along the spatial and channel dimensions, respectively.

### 3.3. Structure-Aware Smooth Loss Function

Given that current methods’ loss functions often overlook structural information in images, lack spatial structure constraints, and suffer from insufficient supervisory guidance in attention mechanisms, this paper designs a structure-aware smooth loss function 
$L{\mathrm{oss}}_{\mathrm{total}}$
, which consists of smooth loss 
$L{\mathrm{oss}}_{\mathrm{smooth}}$
, curvature loss 
$L{\mathrm{oss}}_{\mathrm{curvature}}$
, and attention supervision loss 
$\;L{\mathrm{oss}}_{\mathrm{attention}}$
, as shown in Equation (27).
(27)
$$L{\mathrm{oss}}_{\mathrm{total}}=\lambda_{1}L{\mathrm{oss}}_{\mathrm{smooth}}+\lambda_{2}L{\mathrm{oss}}_{\mathrm{curvature}}+\lambda_{3}L{\mathrm{oss}}_{\mathrm{attention}}$$
where 
$\lambda_{1}$
,

$\lambda_{2}$
 and

$\lambda_{3}$
 are the weight parameters.

(1)Smoothness Loss

Smoothness loss [37] aims to encourage the model’s output to maintain continuous variations in the local space, preventing the occurrence of high-frequency noise or unnatural discontinuities, as shown in Equation (28).
(28)
$$L_{\mathrm{smooth}}=\frac{1}{N}\sum_{i,j}(\left|{\hat{I}}_{i,j}-{\hat{I}}_{i+1,j}\right|+\left|{\hat{I}}_{i,j}-{\hat{I}}_{i,j+1}\right|)$$

where 
N
 represents the total number of pixels, the loss function penalizes abrupt changes by measuring the gradient differences in the horizontal and vertical directions of the output image 
$\hat{I}$
, thus promoting texture consistency in the result. 
i
 and 
j
 represent the row and column index coordinates of the pixels in image 
$\hat{I}$
, i.e., the coordinates of each pixel in the two-dimensional image 
$\hat{I}$
. The smoothness loss function encourages local continuity and mitigates high-frequency noise by imposing a constraint on the first-order gradient (neighborhood pixel difference) of the reconstructed image. Its physical interpretation is that, in natural scenes, the spectra of a given object surface or adjacent regions generally exhibit a smooth spatial transition. Penalizing the first-order difference is conceptually equivalent to imposing a prior on the lower-order statistics of the data, thereby inhibiting the decoder from producing physically implausible oscillations or noise, even in the absence of external labels.

(2)Curvature Loss

Curvature loss [38] further constrains the smoothness of the image at the level of second-order derivatives. It is primarily used to suppress local “sharp deformations” or “spike-like structures” in the reconstructed image, as shown in Equation (29).
(29)
$$L_{curvature}=\frac{1}{N}\sum_{i,j}(\left|{\hat{I}}_{i+1,j}+{\hat{I}}_{i-1,j}+{\hat{I}}_{i,j+1}+{\hat{I}}_{i,j-1}-4{\hat{I}}_{i,j}\right|)$$


This loss function facilitates create a more natural and smooth transition at structural boundaries in the output image by measuring the differences between each pixel in the image 
$\hat{I}$
 and its four neighboring pixels. Curvature loss penalizes the second-order difference or local curvature anomalies, suppressing abrupt local variations and ensuring natural transitions at boundaries rather than discontinuous jumps. Compared to the smoothness term, the curvature term provides stronger geometric regularization with respect to edge shapes and geometric consistency, thereby aiding the network in maintaining structural fidelity at boundaries without relying on pixel-level labels.

(3)Attention Supervision Loss

To enhance the model’s ability to focus on structural information, an unsupervised attention supervision loss mechanism is introduced. This mechanism guides the model to learn from key regions, thereby assigning higher attention weights to them. The loss function 
$L_{attention}$
 is represented by Equation (30).
(30)
$$L_{attention}=\frac{1}{N}\sum_{i,j}{\left(A_{i,j}-M_{i,j}\right)}^{2}$$
where 
$A_{i,j}$
 is the attention matrix obtained by weighting the attention weights generated by the DA submodule, and 
$M_{i,j}$
is the mask matrix generated by the spatial mask of the ADNM submodule (for detailed structure, see Section 3.1.2 and Section 3.2.1). The attention mask and attention map are both derived from the network module’s own computations. The attention loss is based on the consistency constraint between the attention map generated internally by the network and the spatial mask output by the ADNM module. It guides the network to focus its learning resources (parameter updates) on regions with rich structural information or significant spectral differences.

## 4. Results and Analysis

### 4.1. Environmental Configuration

In the experiment, the hardware environment utilized an NVIDIA GeForce RTX 3090 GPU with 24 GB of memory. The CPU model used was an Intel Core i9-12900KF processor, with 128 GB of memory. The operating system was Ubuntu 20.04 LTS, with Python version 3.7.0 and CUDA version 11.1. The experiment uses the Adam optimizer for model training, with an initial learning rate set to 0.0001. The Adam optimizer parameters include a first-order moment estimate 
$\beta_{1}=0.9$
, a second-order moment estimate 
$\beta_{2}=0.999$
, and a minimum parameter 
$\delta =1\times 10^{-8}$
 to prevent division by zero errors. The learning rate decay strategy employs polynomial decay, with a decay power of 1.5.

### 4.2. Overview of Hyperspectral Datasets

#### 4.2.1. Dataset Introduction

To evaluate the network architecture proposed in this paper, three typical hyperspectral reconstruction datasets, NTIRE 2020, Harvard, and CAVE, were used for extensive experimentation.

(1)NTIRE 2020 Dataset

The NTIRE 2020 dataset [39] was introduced by the PIRM community and contains images of natural scenes and artificial objects. Its purpose is to provide a standardized evaluation platform for hyperspectral image reconstruction algorithms. The NTIRE 2020 Spectral Reconstruction Challenge offers a natural hyperspectral image dataset with 510 images, including 450 training images, 30 validation images, and 30 test images. The spatial resolution of the NTIRE 2020 dataset is 512 × 482, consisting of 31 spectral bands, with wavelengths ranging from 400 nm to 700 nm, in increments of 10 nm. The dataset primarily serves as a challenge for evaluating the spectral recovery capabilities of models on natural images. Typical hyperspectral images, such as ARAD_HS_0451 and ARAD_HS_0453, are visualized by selecting images from the 400 nm, 500 nm, 600 nm, and 700 nm bands, as shown in Figure 9.

(2)Harvard Dataset

The Harvard dataset [40] was created by Harvard University, with images captured in natural environments that reflect spectral variations in real-world scenes. This dataset contains 50 hyperspectral images, with 43 images used as the training set and 7 images used as the validation set. The image wavelengths range from 420 nm to 720 nm, with band spacing of 10 nm, totaling 31 spectral bands. The spatial resolution of each image is 1392 × 1040, providing higher spatial detail compared to other datasets. The Harvard dataset is primarily used to test the model’s reconstruction performance in higher resolution and more complex scenes. Hyperspectral images from the dataset, such as img1 and imageb0, are visualized at 400 nm, 500 nm, 600 nm, and 700 nm bands, as shown in Figure 10.

(3)CAVE Dataset

The CAVE dataset [41] was created by the laboratory at Columbia University, covering a wide range of object and scene types, including food, toys, fabric, and more. These images were captured in a controlled laboratory environment to eliminate the influence of ambient light on image quality. This dataset contains 32 hyperspectral images, with the first 28 images used as the training set and the last 4 images used as the validation set. The images have a spatial resolution of 512 × 512, and each image contains 31 spectral bands with wavelengths ranging from 400 nm to 700 nm, spaced 10 nm apart. Each spectral band image is a 16-bit grayscale image, ensuring the precision and rich detail of the hyperspectral images. The CAVE dataset is used to evaluate the model’s spectral reconstruction capability in controlled environments. Hyperspectral images from the dataset, such as balloons_ms and fake_and_real_food_ms, are visualized at the 400 nm, 500 nm, 600 nm, and 700 nm bands, as shown in Figure 11.

#### 4.2.2. Evaluation Metrics

When comparing the performance of different spectral reconstruction methods, it is necessary to have standardized metrics for objective evaluation and analysis. Common spectral reconstruction evaluation metrics include Mean Relative Absolute Error (MRAE), Root Mean Square Error (RMSE), and Peak Signal-to-Noise Ratio (PSNR) [39].

MRAE measures the average relative error between the predicted and true values. It is highly sensitive to regions with low reflectance values and reflects the model’s robustness in low-light or dark areas. Its formula is represented by Equation (31).
(31)
$$MRAE=\frac{1}{N}\sum_{i=1}^{N}\frac{\left|H_{i}-{\hat{H}}_{i}\right|}{H_{i}}$$


RMSE measures the overall deviation between the predicted image and the true image, representing the square root of the sum of squared errors, as shown in Equation (32). It reflects the intensity of the global error.
(32)
$$RMSE=\sqrt{\frac{1}{N}\sum_{i=1}^{N}{\left(H_{i}-{\hat{H}}_{i}\right)}^{2}}$$


PSNR measures the maximum difference between the predicted image and the original image and is commonly used to evaluate image reconstruction and compression quality. Its formula is represented by Equation (33).
(33)
$$PSNR=10\cdot \log_{10}(\frac{MAX^{2}}{MSE})$$


In Equations (31)–(33), 
$H_{i}$
 and 
${\hat{H}}_{i}$
 represent the 
i
 pixel in the true image and the reconstructed image, respectively. 
N
 denotes the total number of pixels in both the true and reconstructed images. 
MAX
 is the maximum pixel value, for example, if each sample point is represented using 8 bits, then 
MAX
 is 255.

Generally, the smaller the MRAE and RMSE, and the larger the PSNR, the better the performance of the image reconstruction model.

### 4.3. Comparison and Analysis of Experimental Results

To better demonstrate the effectiveness of the proposed DMMAF network model, we compared DMMAF with four advanced SOTA methods: HRNet [42], AWAN [43], MFormer [18] and GMSR [44]. HRNet and AWAN ranked first in the NTIRE 2020 challenge’s Real World and Clean tracks, respectively. MFormer demonstrated outstanding performance under unsupervised learning across multiple datasets, while GMSR, which incorporates the Mamba framework, exhibited promising and efficient reconstruction results. It is worth noting that MFormer is an unsupervised training algorithm, while the other methods are supervised training algorithms. To ensure a fair comparison, the supervised training algorithms were modified to use the same unsupervised training process, enabling a comparison of reconstruction performance under identical conditions.

#### 4.3.1. Comparison and Analysis of Visualization Results

To better showcase the reconstruction performance of our algorithm and other methods across different spectral bands, we selected the same two bands from the 31 spectral bands of the ARAD_HS_0453, balloons_ms, and imgf7 hyperspectral images from the NTIRE 2020-CLEAN, Harvard, and CAVE datasets. The corresponding reconstructed HSI and real HSI are displayed using pseudocolor for visualization, as shown in Figure 12, Figure 13 and Figure 14.

For the 450 nm and 680 nm band images of ARAD_HS_0453 from the NTIRE2020-CLEAN dataset, our algorithm outperforms other methods in terms of edge detail richness and robustness, as indicated by the red boxes in Figure 12. In the 450 nm band image marked by the red box in the second row, the image reconstructed by the DMMAF algorithm has fewer edge artifacts and more detailed features.

For the 450 nm and 680 nm band images of imgf7 in the Harvard dataset, as indicated by the red boxes in Figure 13, DMMAF shows higher attention to key details and achieves better image reconstruction accuracy compared to other methods. In the 450 nm band image marked by the red box in the second row, and in the 680 nm band image marked by the red box in the third row, DMMAF’s local and global modeling of the image is closer to the ground truth image compared to other algorithms, with a more complete representation of details.

For the 450 nm and 680 nm band images of balloons_ms from the CAVE dataset, as indicated by the red boxes in Figure 14, DMMAF demonstrates superior spectral resolution and spatial resolution balance in image reconstruction compared to other methods. When the spectral resolution is high but the spatial resolution is low, the image details become blurry. Conversely, when the spatial resolution is high but the spectral resolution is low, the color variations in the image are minimal or appear too coarse. In the 450 nm and 680 nm band images marked by the red boxes in the second and third rows, DMMAF shows the smallest difference in image details and color variations compared to the ground truth image, showcasing its excellent spatial resolution balance.

Figure 15 Displays the spectral response curves of the reconstructed HSI, and our DMMAF is closer than other methods to the ground truth intuitively.

#### 4.3.2. Quantitative Results Comparison and Analysis

The HRNet, AWAN, MFormer, GMSR, and DMMAF framework models were retrained on the same hardware, programming environment, and dataset to ensure a fair comparison. They were evaluated using the MRAE and RMSE metrics on the same validation set. Due to the presence of zero spectral values in the CAVE and Harvard datasets, MRAE could not be computed; thus, the evaluation metrics were limited to RMSE and PSNR. The final quantitative results for these evaluation algorithms are presented in Table 1.

On the NTIRE2020-CLEAN dataset, our algorithm outperforms the state-of-the-art GMSR algorithm by 2.2% in terms of MRAE, achieving a value of 0.133. Compared to the state-of-the-art MFormer algorithm, our method reduces RMSE by 5.0% and increases PSNR by 0.15%, achieving values of 0.040 and 31.314, respectively. On the CAVE dataset, RMSE is reduced by 2.4% compared to the best-performing GMSR algorithm, reaching 0.041. PSNR improves by 1.7% compared to MFormer, reaching 30.983. On the Harvard dataset, compared to MFormer, RMSE is reduced by 3.8%, reaching 0.025, and PSNR increases by 0.7%, reaching 34.955.

#### 4.3.3. Ablation Study

Due to the NTIRE2020-CLEAN dataset containing more image samples and covering a wider variety of scene types compared to other datasets, it was selected for the ablation study. The evaluation metrics used for this dataset are PSNR and RMSE.

To demonstrate the effectiveness of the ADNM module, DA module, and DGM module, relevant ablation experiments were conducted. The final quantitative results of the ablation experiments are shown in Table 2.

The NTIRE2020-CLEAN dataset contains a larger number of image samples compared to other datasets, covering a wider range of scene types. Therefore, this dataset was selected for the ablation studies, with PSNR and RMSE used as the evaluation metrics.

The quantitative metrics of PSNR and RMSE when the ADNM, DA, and DGM submodules are removed from the DMMAF network are presented in Table 2. Compared to the complete DMMAF network, the removal of the ADNM, DA, or DGM submodule results in a decrease in PSNR by 0.8%, 2.0%, and 6.6%, respectively, and an increase in RMSE by 11.1%, 18.3%, and 24.2%. Therefore, it can be inferred that the performance of the DMMAF network with all modules intact significantly outperforms that of any configuration with missing submodules.

In particular, to showcase the contribution of each individual component to the model’s performance, we incrementally added the Adaptive Dual-Noise-Aware Feature Extraction Module (ADNM), the Deformable Attention (DA) Module, and the Dual-Gated Mamba Multi-Scale Module (DGM) to the baseline model, which only includes the RDPFE and the basic structure for reconstructing images.

As shown in Table 3, each component makes a significant contribution to the performance. Adding the ADNM module to the BASE model improved the PSNR by 6.659 and reduced the RMSE by 0.245. This indicates that ADNM plays an active role in denoising and detail recovery. Adding the DA module to this combination resulted in a PSNR improvement of 9.411 and an RMSE reduction of 0.057, demonstrating that the DA module effectively enhances the attention to global features, further improving the quality of the reconstructed image. Finally, adding the DGM module to the new combination increased the PSNR by 2.052 and reduced the RMSE by 0.013, highlighting the significant role of the DGM module in optimizing local feature modeling and reducing redundant parameters.

Meanwhile, to visually demonstrate the effectiveness of the three modules in the network architecture, we conducted experiments by eliminating the ADNM, DA, and DGM modules from the complete model (as shown in Table 2) and visualized the results. In the ablation experiment, the NTIRE2020-CLEAN typical example dataset ARAD_HS_0463 was selected for visual comparison. The pseudocolor visualizations of the selected five bands are shown in Figure 16. The first, second, and third rows represent the results without the DGM, DA, and ADNM submodules, respectively, while the fourth row shows the true hyperspectral image.


w/o
 ADNM, 
w/o
 DA and 
w/o
 DGM represent the DMMAF network with the corresponding modules missing, while the remaining modules are fully intact. Regions of the image with significant differences compared to the ground truth are highlighted with black boxes for better visual comparison. When the DMMAF network lacks the ADNM module, the reconstruction in the 480 nm, 540 nm, and 700 nm bands exhibits issues such as detail loss and image noise. When the DMMAF network lacks the DA module, the reconstruction in the 480 nm, 540 nm, and 700 nm bands shows blurred high-frequency details in local regions. When the DMMAF network lacks the DGM module, the reconstruction in the 480 nm, 540 nm, and 700 nm bands presents rough global low-frequency details that do not match the color distribution of the real scene. This demonstrates that the ADNM, DA, and DGM modules play an indispensable role in noise handling, capturing local detail information, and utilizing global information interactions.

We also performed model training with round-by-round loss on the NTIRE2020-CLEAN dataset and plotted the PSNR values for each epoch. The chart below shows the PSNR values achieved at each epoch. As seen in the graph, the model reached the best PSNR value (31.314) at the 100th epoch, as shown in Figure 17.

Although the PSNR values fluctuate slightly in some epochs, they generally show a stable trend as training progresses. DMMAF reaches its optimal PSNR value of 31.314 at the 100th epoch, which slightly drops to 29.983 at the 200th epoch, as shown in Table 4.

To verify that the adaptive dual-noise masking schedule (
p
, 
r
) can improve stability, we supplemented the PSNR metrics for different epochs of masking effects on the NTIRE2020 dataset, as shown in Table 5. Since the masking schedule ratio changes continuously with the epochs, we selected the corresponding masking ratios and PSNR results for epochs = 1, 25, 50, 75, and 100. To further validate its effectiveness, we also conducted a simple random masking experiment (simpler stochastic dropout). In this experiment, a fixed masking ratio of 0.3 was set, and training was conducted at epoch = 100.

As the model progresses through more epochs, both the image masking ratios 
p
 and 
r
, along with their corresponding PSNR values, gradually increase, as shown in Figure 18. The PSNR value reached its optimal value of 31.045 at epoch = 100, which is an improvement of 1.03 compared to the simple masking result of 30.015.

## 5. Discussion

This study presents the Dual-Gated Mamba Multi-Scale Adaptive Feature Learning (DMMAF) network, which demonstrates exceptional performance in unsupervised hyperspectral image reconstruction from a single RGB image, particularly in terms of image naturalness, detail accuracy, and model robustness, showing significant improvements over existing methods.

Our proposed DMMAF utilizes an unsupervised learning approach, achieving hyperspectral image reconstruction without relying on labeled information. The loss function in this study consists of three parts: smoothness loss (Equation (28)), curvature loss (Equation (29)), and attention supervision loss (Equation (30)). These losses do not provide training signals through pixel-level comparisons between the reconstructed HSI and the true HSI or RGB images, but rather they provide self-constraints to the network based on low-order statistics and structural consistency of the reconstructed results. The smoothness/curvature terms penalize low-order statistical biases and second-order geometric anomalies, acting as “weak-label” priors (without using any external labels). The attention term further reduces the uncertainty in self-supervised training by enhancing the internal consistency of the representations. These unsupervised constraints collectively replace the pixel-level guidance provided by explicit labels, enabling the model to learn stable and physically meaningful spectral mappings without relying on paired HSI labels.

In terms of visual image reconstruction quality, we performed a cross-method comparative analysis. The DMMAF method demonstrates outstanding performance in both image naturalness and detail accuracy, outperforming existing methods, including HRNet [42], AWAN [43], MFormer [18], and GMSR [44], four advanced SOTA methods. DMMAF designed a feature extraction method combining RDPFE and ADNM, which improves edge detail richness and robustness in the reconstructed image, as shown in Figure 12. This is due to the RDPFE module, which combines depth convolution for spatial and channel adjustment with pixel-level convolution, effectively reducing edge artifacts and retaining edge details closer to the true image. The ADNM module’s adaptive masking strategy allows the model to autonomously learn relevant features, leading to smoother color transitions in these regions and better matching the true image. DMMAF also constructs the DA and DGM modules for local–global feature association modeling, which improves the attention to key details and accuracy in the reconstructed image, as shown in Figure 13. This is mainly due to the DA module’s flexibility in capturing attention areas and adapting to object shapes, enhancing the global detail integrity of the reconstructed image. The DGM module’s precise control of long-range dependency modeling improves the accuracy of local feature representations. Furthermore, DMMAF uses a structure-aware smoothness loss function to balance spectral and spatial resolution, as shown in Figure 14. This is because the structure-aware smoothness loss function effectively addresses multiple issues, including information neglect, insufficient spatial structure constraints, and lack of attention supervision. As a result, the reconstructed image shows smoother transitions in structure and color, along with more detailed features. Compared to the true hyperspectral image, the reconstructed images from DMMAF show better performance in terms of certain details and boundaries, although there is still a slight gap. This is mainly related to the dataset quality, input image resolution, and prior assumptions during model training. Nevertheless, DMMAF’s reconstructed images demonstrate high spectral consistency and structural fidelity in most scenarios, making it an effective approximation of hyperspectral images. The overall visualization comparison results indicate that DMMAF excels in recovering object boundaries, details, and textures, demonstrating good practicality. In applications such as remote sensing imaging, precision irrigation in agriculture, and ecological environment monitoring, hyperspectral images provide rich spectral information, helping researchers obtain more accurate surface reflectance data. The outputs from DMMAF can provide useful hyperspectral data for these application scenarios without the need for expensive hyperspectral sensors, increasing the practicality of this method.

DMMAF also demonstrates the highest reconstruction accuracy through quantitative comparisons with HRNet, AWAN, MFormer, and GMSR, as shown in Table 1. DMMAF outperforms state-of-the-art unsupervised hyperspectral reconstruction algorithms across all three datasets, effectively recovering details and maintaining high structural consistency. While the AWAN method shows good performance in hyperspectral image reconstruction, it has certain flaws in detail recovery and noise suppression. In contrast, DMMAF compensates for information loss and suppresses noise through the ADNM module’s masking mechanism, improving model robustness. HRNet, based on a high-resolution network, excels in processing high-resolution images, but its performance in balancing spectral information and detail recovery is limited by its model structure. DMMAF addresses this issue with the structure-aware smoothness loss function, achieving better results in spectral texture and structural fidelity. The GMSR method, based on image restoration technology, performs well in image quality but lacks global information modeling capability. DMMAF, under the influence of the DA module, outperforms GMSR in this regard, significantly improving image reconstruction accuracy. MFormer, a transformer-based model, has strong global information modeling capability, but may suffer from information loss when handling local details and complex scenes. DMMAF overcomes this limitation through deep dot-product feature extraction and double Mamba feature extraction, showing particular advantages in complex boundaries. However, in this study, only the NTIRE2020, Harvard, and CAVE datasets with 31 similar bands were used. In the future, we plan to use datasets with different bands, especially for applications in remote sensing and medical imaging, to enhance the model’s adaptability and portability. Additionally, the current evaluation metrics are primarily based on PSNR, RMSE, and MRSE, without involving additional metrics such as SAM and SSIM, which could offer a more objective and comprehensive evaluation of the algorithm’s effectiveness.

This study conducted two ablation experiments: (1) eliminating the ADNM, DA, and DGM modules from the complete model, as shown in Table 2, and (2) sequentially adding the ADNM, DA, and DGM modules to the baseline model, as shown in Table 3. The data from Table 2 shows that when the ADNM module’s real-time masking function is removed, the model’s ability to handle noise is reduced. When the DA and DGM submodules are removed, the network’s ability to capture local features and build global features is compromised. These effects are visually demonstrated in the results shown in Figure 16, indicating the indispensable role of the DGM, DA, and ADNM modules in capturing local detail information, handling noise, and utilizing information interactions. The experimental quantification results in Table 3 clearly show that the sequential addition of each module significantly enhances the overall model performance, further proving the importance of each module in improving hyperspectral image reconstruction quality, particularly in detail preservation, denoising, and global feature modeling. However, ablation experiments across datasets have not yet been conducted, so it is uncertain whether the approach is effective across different data distributions.

DMMAF conducted two experiments: one with epoch-by-epoch loss training and the other with the masking ratio varying epoch by epoch, as detailed in Table 4 and Table 5. Due to the structure-aware smoothness loss function, DMMAF exhibits relatively stable performance during most epochs, as shown in Figure 17. When the training epochs are too few (less than 100), the model cannot fully understand the mapping between images, leading to poor reconstruction performance. When the training epochs are too many (greater than 100), overfitting occurs, resulting in poor generalization. In the visual results shown in Figure 18, the model reaches its highest performance at epoch = 100. This is because, under the influence of the ADNM module, the model’s masking ratio increases as training progresses, forcing the model to repeatedly relearn spatial and channel structure information adaptively.

## 6. Conclusions

This study presents the DMMAF network framework for unsupervised hyperspectral image (HSI) reconstruction from a single RGB image. The main innovation of DMMAF lies in its dual-gated Mamba multi-scale adaptive feature learning paradigm, which tightly integrates a noise-aware edge detail extractor based on RDPADN, a deformable attention dual-gated Mamba module for joint local–global modeling, and a structure-aware smoothness loss function that provides unsupervised spatial and spectral guidance. DMMAF enhances the preservation of spatial structure and high-frequency spectral details and improves robustness to noise and illumination changes through adaptive masking and dual-gated feature modulation while maintaining reasonable computational complexity. This results in superior reconstruction accuracy compared to existing state-of-the-art supervised and unsupervised methods. Compared to the best-performing unsupervised algorithms, DMMAF improves PSNR, RMSE, and MRAE by 0.15%, 5.0%, and 2.2%, respectively, on the NTIRE2020 dataset. On the Harvard and CAVE datasets, DMMAF improves PSNR and RMSE by 0.7%, 3.8% and 1.7%, 2.4%, respectively. Extensive experimental results, both numerical and visual, demonstrate that our algorithm effectively addresses issues such as detail loss, insufficient robustness, low reconstruction accuracy, and the difficulty of achieving a balance between spatial and spectral resolution, highlighting the superiority and practical application potential of DMMAF. In the future, we will expand the evaluation metrics, use datasets with different bands, and conduct cross-dataset experiments to further demonstrate the effectiveness and generalization of the DMMAF algorithm.

## Figures and Tables

**Figure 1 jimaging-12-00019-f001:**
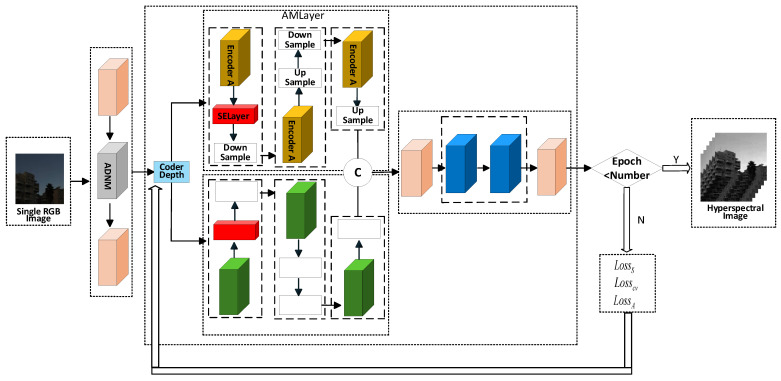
An overview of the proposed DMMAF network and illustrates the interactions among its three core components: (**a**) Reflection Dot-product Adaptive Dual-noise-aware Feature Extraction (RDPADN), (**b**) Deformable Attention Dual-Gated Mamba Multi-Scale Feature Learning (DADGM), and (**c**) Structure-Aware Smooth Constraint Loss Function.

**Figure 2 jimaging-12-00019-f002:**
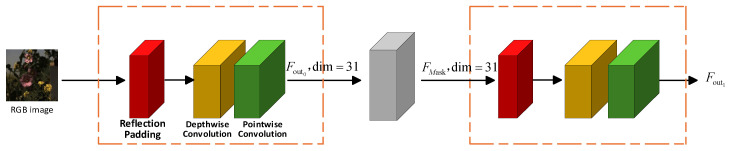
Reflection Dot-Product Adaptive Dual-Noise-Aware Feature Extraction.

**Figure 3 jimaging-12-00019-f003:**
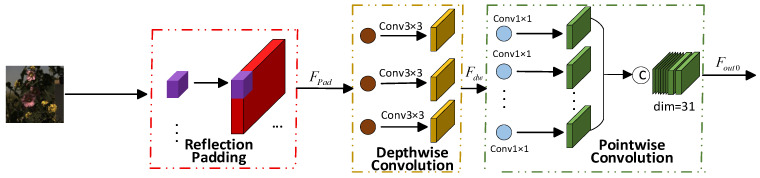
Reflection Depth Point Feature Extraction.

**Figure 4 jimaging-12-00019-f004:**
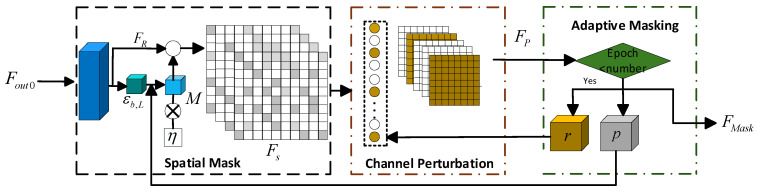
Adaptive Dual-Noise Masking.

**Figure 5 jimaging-12-00019-f005:**
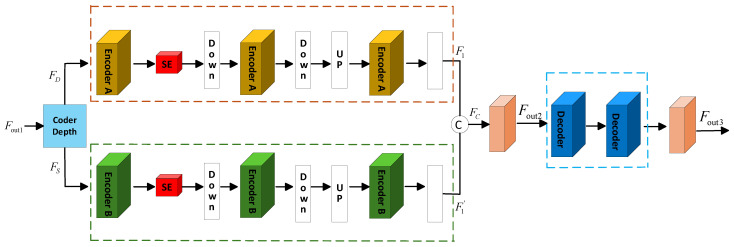
Deformable Attention Dual-Gated Mamba Multi-Scale Feature Learning Process.

**Figure 6 jimaging-12-00019-f006:**
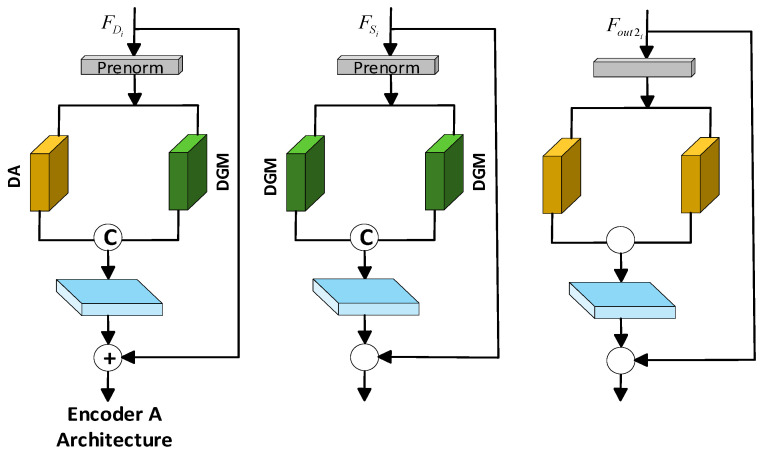
Encoder–Decoder Structure.

**Figure 7 jimaging-12-00019-f007:**
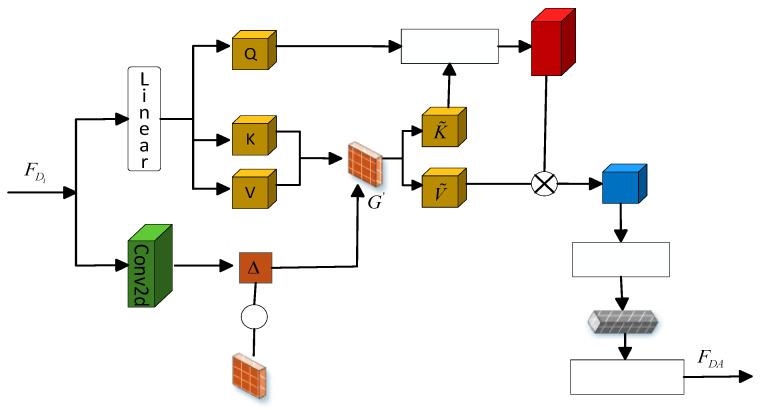
Dynamic Adaptive Deformable Attention Global Feature Extraction.

**Figure 8 jimaging-12-00019-f008:**
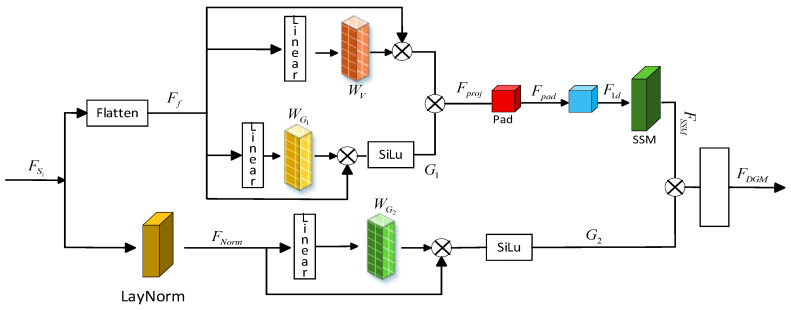
Dual-Gated Mamba Local Feature Extraction.

**Figure 9 jimaging-12-00019-f009:**
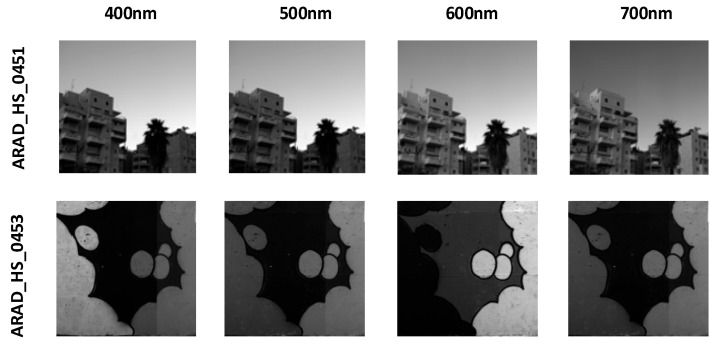
Visualization of typical hyperspectral images ARAD_HS_0451 and ARAD_HS_0453 from the NTIRE 2020 dataset at different wavelengths.

**Figure 10 jimaging-12-00019-f010:**
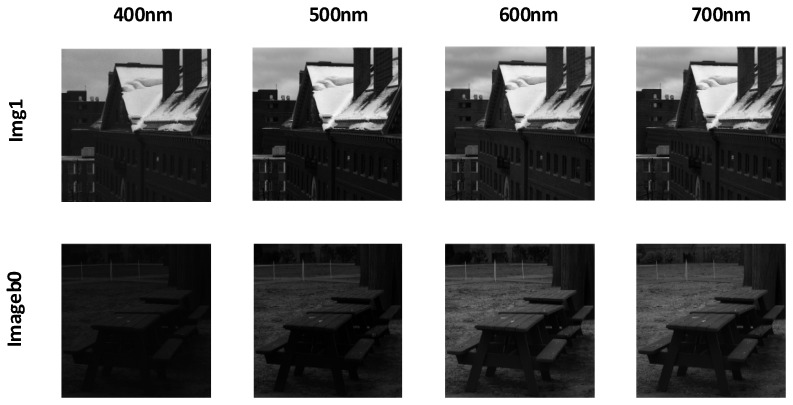
Visualization of typical hyperspectral images img1 and imageb0 from the Harvard dataset at different wavelengths.

**Figure 11 jimaging-12-00019-f011:**
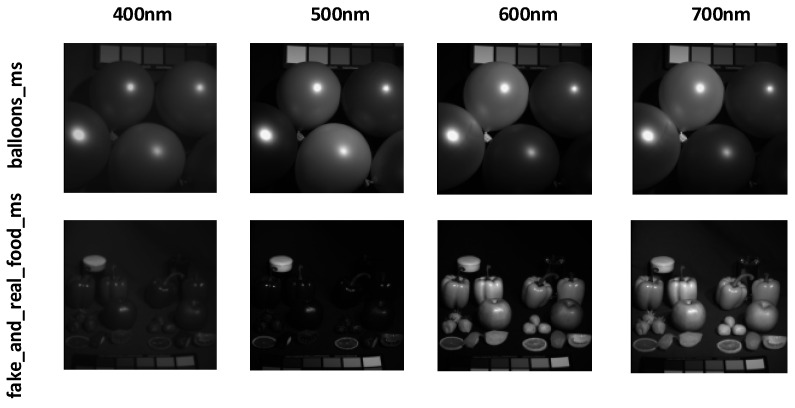
Visualization of typical hyperspectral images balloons_ms and fake_and_real_foods_ms from the CAVE dataset at different wavelengths.

**Figure 12 jimaging-12-00019-f012:**
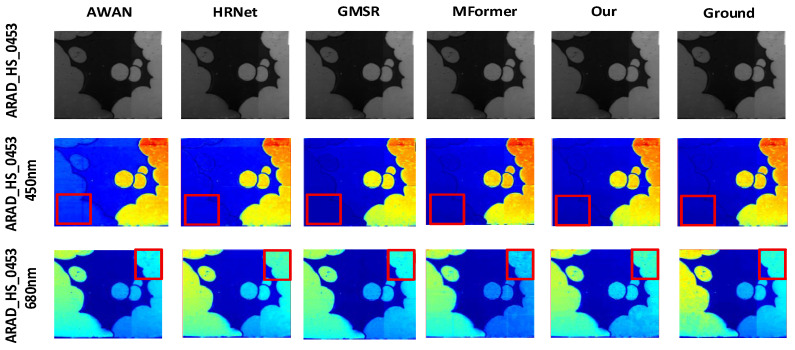
Pseudocolor Effect of Reconstructed Hyperspectral Image for ARAD_HS_0453 under Different Models.

**Figure 13 jimaging-12-00019-f013:**
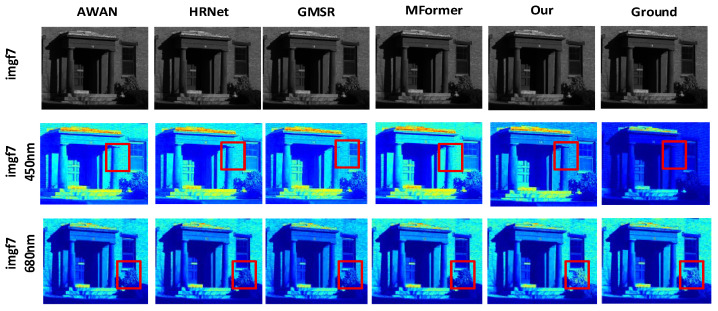
Pseudocolor Effect of Reconstructed Hyperspectral Image for imgf7 under Different Models.

**Figure 14 jimaging-12-00019-f014:**
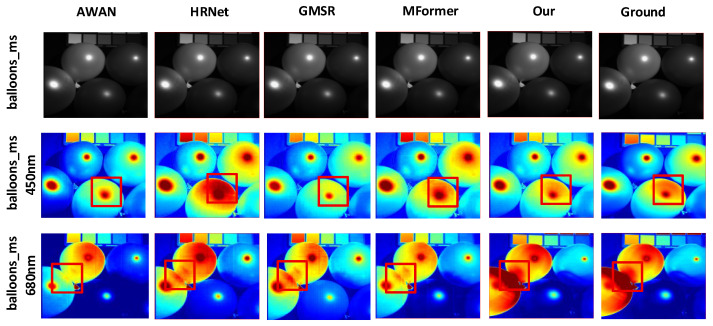
Pseudocolor Effect of Reconstructed Hyperspectral Image for balloons_ms under Different Models.

**Figure 15 jimaging-12-00019-f015:**
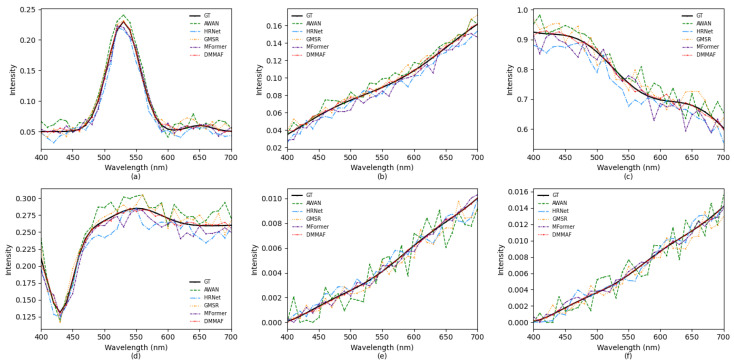
Spectral response curves of randomly selected several spatial points from the reconstructed HSI of each SOTA method and the ground truth HSI. (**a**,**b**) for the NTIRE2020-Clean. (**c**,**d**) for the CAVE. (**e**,**f**) for the Harvard.

**Figure 16 jimaging-12-00019-f016:**
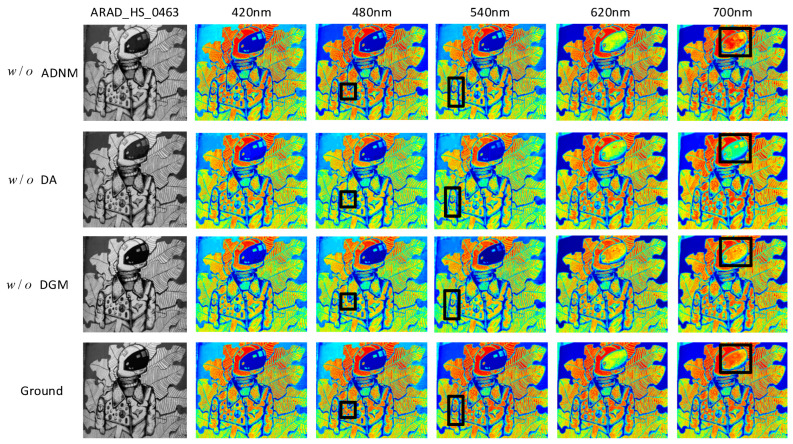
Pseudocolor Visualization of ARAD_HS_0463.

**Figure 17 jimaging-12-00019-f017:**
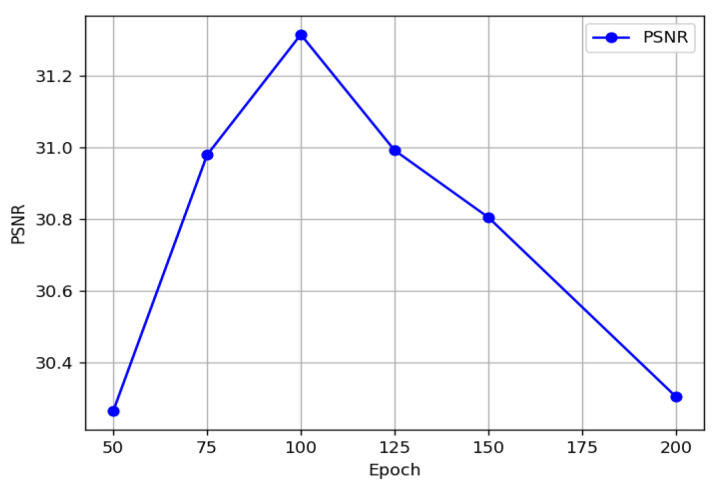
Epoch-by-Epoch PSNR.

**Figure 18 jimaging-12-00019-f018:**
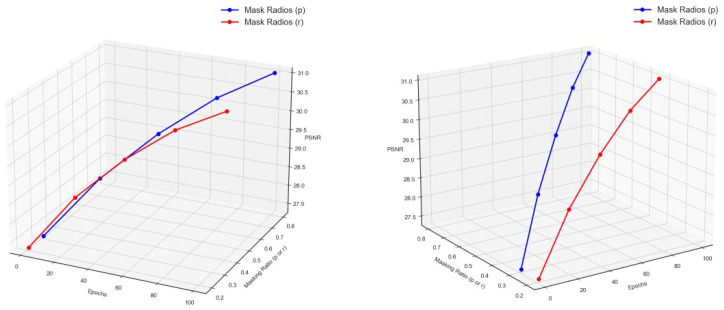
Visualization of PSNR metric changes across training epochs for different Mask Ratios.

**Table 1 jimaging-12-00019-t001:** Final Test Results for NTIRE2020-CLEAN, CAVE, and HARVARD Datasets.

Method	NTIRE2020-CLEAN			CAVE		Harvard	
	MRAE (↓)	RMSE (↓)	PSNR (↓)	RMSE (↓)	PSNR (↓)	RMSE (↓)	PSNR (↓)
MFormer [18]	0.139	0.042	31.265	0.044	30.467	0.026	34.697
HRNet [42]	0.256	0.058	27.436	0.054	27.865	0.041	32.143
AWAN [43]	0.176	0.052	29.192	0.051	28.486	0.035	32.861
GMSR [44]	0.136	0.043	30.822	0.042	29.995	0.031	33.951
DMMAF	**0.133**	**0.040**	**31.314**	**0.041**	**30.983**	**0.025**	**34.955**

**Table 2 jimaging-12-00019-t002:** Quantitative Comparison of Ablation Studies (1) on the NTIRE2020-CLEAN Dataset.

MODULE	w/o ADNM	w/o DA	w/o DGM	Our
PSNR ( ↑ )	31.062	30.679	29.262	**31.314**
RMSE ( ↓ )	0.045	0.049	0.053	**0.040**

**Table 3 jimaging-12-00019-t003:** Quantitative Comparison of Ablation Studies (2) on the NTIRE2020-CLEAN Dataset.

MODULE	BASE	BASE + ADNM	BASE + ADNM + DA	BASE + ADNM + DA + DGM
PSNR ( ↑ )	13.192	19.851	29.262	31.314
RMSE ( ↓ )	0.355	0.110	0.053	0.040

**Table 4 jimaging-12-00019-t004:** PSNR values for epoch-by-epoch loss training on the NTIRE2020-CLEAN dataset.

Dataset	NTIRE2020-CLEAN					
Epoch	50	75	100	125	150	200
PSNR ( ↑ )	30.264	30.978	31.314	30.991	30.804	30.304

**Table 5 jimaging-12-00019-t005:** PSNR metric for masking performance at different epochs on the NTIRE2020 dataset.

Method	Adaptive Dual-Noise Mask					Mask
Epoch	1	25	50	75	100	100
Mask_Ratio(p)	0.3	0.425	0.55	0.675	0.8	0.3
Mask_Ratio(r)	0.2	0.25	0.3	0.35	0.4	0.3
PSNR ( ↑ )	27.324	28.689	29.722	30.521	31.045	30.015

## Data Availability

NTIRE 2020 [39], Harvard [40] and CAVE [41] datasets used in this study are publicly available at https://data.vision.ee.ethz.ch/cvl/ntire20/ (accessed on 10 October 2025), https://vision.seas.harvard.edu/hyperspec/download.html (accessed on 10 October 2025) and https://cave.cs.columbia.edu/repository/Multispectral (accessed on 8 October 2025). Source code can be downloaded at https://github.com/Zzz2333333/DMMAF (accessed on 20 October 2025).

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
