# Peer review of "Double-Gated Mamba Multi-Scale Adaptive Feature Learning Network for Unsupervised Single RGB Image Hyperspectral Image Reconstruction"

_2313-433X, 2025, doi:10.3390/jimaging12010019_

Round 1
Reviewer 1 Report
Comments and Suggestions for Authors
The article appears impressive overall. It appears to be a strong contribution to the literature. However, I believe some sections need to be edited, clarified, and structurally improved.
I recommend the following revisions for this article:
1. The article is impressive and makes a strong contribution to the literature. However, the model's origins should be more clearly emphasized; the extensive literature in the introduction should be moved to the "Related Works" section. This will make the model's motivation and main contribution clearer.
2. It would be helpful to present epoch-by-epoch loss and performance graphs to understand the model's training process.
3. The model's performance can be analyzed in more detail. The usefulness of the resulting outputs, and whether they can truly be considered HSI images, should be discussed.
4. The model contains numerous components. The contribution of these components to performance should be quantified.
Author Response
请查看附件。

Reviewer 2 Report
Comments and Suggestions for Authors
Dear Authors,
First of all, I congratulate you for your work. I have completed my evaluation of your work. I have indicated below the areas that I see necessary in the study. I hope that making the revisions specified in these items will contribute to your work. I wish you success.
Evaluation:
• It would be useful to mention the metric results in the summary section. It would also be beneficial to briefly mention the research findings and contributions in 1-2 sentences in the summary section. • The study contains a certain number of punctuation and spelling errors. It would be beneficial to review the entire study. • In Section 3.1, line 568, it says [value missing]. • Figure 8 is not cited in the text. • The study does not have a discussion section. The discussion section attempts to present the results, but it is insufficient. Also, what are the limitations of the study? These should be mentioned. Similarly, a comparative discussion with similar studies conducted using the same dataset would contribute to the study.Author Response
Please see the attachment

Reviewer 3 Report
Comments and Suggestions for Authors
The abstract reads like a methods list and SOTA claim without quantitative summaries or caveats. It asserts the loss “effectively resolves the spatial–spectral trade-off” and that results are “state-of-the-art” without reporting effect sizes or statistical variability.
Several background statements are either imprecise or contain typographical/formatting errors (e.g., “The spectral resolution of HSI is on the order of 2 10 λ−” — unclear notation/units). References are densely enumerated but the narrative doesn’t identify the precise failure modes of prior unsupervised RGB→HSI models that your method addresses.
Notation is inconsistent (RDPADN vs RDPFE; DADGM vs the text’s “Deformable Attention Dual-Gated Mamba”—sometimes “offset attention”). This makes the execution path difficult to follow and hinders reproducibility.
No justification is given for choosing 31 (matching NTIRE bands) when later you also evaluate Harvard/CAVE; how is band-mismatch handled?
The “adaptive dual-noise masking” schedules (ρ, r) are asserted to improve stability, but there’s no sensitivity analysis, no visualization of masks across epochs, and no comparison to simpler stochastic dropout.
The odd/even layer split into AMLayer/DMLayer is an unusual design choice; however, you don’t justify why parity-based routing (vs learned gating) improves multi-scale fusion. The path description (down–down–up–up) is narrative; there’s no algorithmic listing or tensor shape table.
The DGM description mixes 1D conv, padding, and SSM kernels but omits sequence length definition and state size; there’s no clarity on how 2D features are linearized and whether scan is causal/non-causal. Gating functions (G1, G2) are asserted to modulate spatial and channel dimensions, but the equations do not demonstrate explicit spatial conditioning beyond a flatten step.
Critical hyperparameters are missing or inconsistent: the initial learning rate is “[value missing]”; decay schedule is given without optimizer step mapping; 94,500 “max iteration step” appears arbitrary without dataset/patch count linkage.
Descriptions are lengthy but do not state train/val/test splits used by you (esp. for Harvard/CAVE) or any pre-processing (RGB formation from HSI, camera response modeling, patching, augmentation). For NTIRE, it’s unclear how you handled the challenge’s test set restrictions in an unsupervised setting.
You omit widely used spectral metrics such as SAM and ΔE, and spatial metrics like SSIM; relying mainly on RMSE/PSNR hides spectral fidelity errors. You also state MRAE cannot be computed on Harvard/CAVE due to zeros but provide no mitigation.
you mentioned supervised baselines were “modified to use the same unsupervised training process” but do not detail how losses/regularizers/CRFs were removed or replaced. Without code/flags for baselines, the SOTA claim is weak.
Module ablations are coarse (drop entire ADNM/DA/DGM) and reported only on NTIRE-CLEAN with two metrics. No cross-dataset ablations, no learning-curve or mask-schedule sensitivity, and no visualization of spectral error curves.
Reviewer 4 Report
Comments and Suggestions for Authors
- Abstract is need to improve with qualitative results of the research
- Introduction section is very length - Reduce as the research important
- Add Novelty, aim, objective and scope of the research in the last para of the manuscript.
- Figure 1 need more explanation
- Equation explanation is missing throughout the manuscript
- What is the main difference you are found in the manuscript?
- Very less explanation found in the fig no 10 - Need more explanation
- What is the main difference in table 1
- Conclusion is very less - add quantitative results
Reviewer 5 Report
Comments and Suggestions for Authors
Please see the attached file.

Author Response
请查看附件。

Round 2
Reviewer 1 Report
Comments and Suggestions for Authors
The authors have generally addressed the concerns I raised about the article and made the necessary corrections. I recommend accepting the article for publication.
Reviewer 3 Report
Comments and Suggestions for Authors
I appreciate the authors’ efforts in addressing my comments and suggestions to improve the quality of their paper.